# Novel Features for Binary Time Series Based on Branch Length Similarity Entropy

**DOI:** 10.3390/e23040480

**Published:** 2021-04-18

**Authors:** Sang-Hee Lee, Cheol-Min Park

**Affiliations:** Division of Industrial Mathematics, National Institute for Mathematical Sciences, Daejeon 34047, Korea; mpcm@nims.re.kr

**Keywords:** binary time-series, branch length similarity entropy, time circle, binarization

## Abstract

Branch length similarity (BLS) entropy is defined in a network consisting of a single node and branches. In this study, we mapped the binary time-series signal to the circumference of the time circle so that the BLS entropy can be calculated for the binary time-series. We obtained the BLS entropy values for “1” signals on the time circle. The set of values are the BLS entropy profile. We selected the local maximum (minimum) point, slope, and inflection point of the entropy profile as the characteristic features of the binary time-series and investigated and explored their significance. The local maximum (minimum) point indicates the time at which the rate of change in the signal density becomes zero. The slope and inflection points correspond to the degree of change in the signal density and the time at which the signal density changes occur, respectively. Moreover, we show that the characteristic features can be widely used in binary time-series analysis by characterizing the movement trajectory of Caenorhabditis elegans. We also mention the problems that need to be explored mathematically in relation to the features and propose candidates for additional features based on the BLS entropy profile.

## 1. Introduction

Time series is a ubiquitous and widely used data type owing to the prevalence of Internet-based network information. Data are generated in several fields, including medicine and healthcare, science, finance, economics, government, industry, environmental science, and socio-economics [1,2,3,4,5,6]. Thus, over the past decade, researchers have developed various approaches to analyze data to understand the properties of various systems in diverse fields [7,8,9,10]. The purpose of analysis is primarily to predict signal occurrence, classify time series into one or several classes, detect anomalies or motifs contained in the data [11], or quantify similarities (or dissimilarities) between time series [12,13]. The approaches can be classified into four categories depending on the purpose.

Distance-based approaches use the similarity between time series. Among them, Euclidean distance (ED) measurements and dynamic time distortion (DTW) measurements are widely used. ED calculates the similarity as the square root of the sum of squared differences between elements corresponding to the same time position in two time series with the same length. While this measurement is simple and intuitive, it has the disadvantage of being overly sensitive to outliers, making it difficult to compare time series of different lengths. DTW, which overcomes the abovementioned drawback, is an effective method of minimizing the effects of motion and distortion over time by allowing the detection of a similar shape out of phase through an “elastic” transformation of the time series [14,15]. Lines and Bagnall compared several distance measures and showed that no distance measure significantly outperforms DTW [16].

Feature-based approaches extract structural features that reflect the properties of the time series and analyze the extracted features using an existing classification method [17]. These methods typically include signal processing techniques using various transforms, such as the discrete Fourier transform (DFT) or discrete wavelet transform (DWT). Most DFT and DWT algorithms are applied to complex patterned time series. The DFT algorithm maps the time series to the frequency domain and represents the time series as a set of basic functions (mainly sine and cosine functions). Owing to its fundamentals, DFT is particularly efficient for evaluating periodic signals. DWT has several basic features and provides multiresolution signal decomposition, which demonstrates an excellent performance for accurate clustering of time series with a high similarity into homogeneous groups [18,19,20,21].

Model-based approaches use a time series generated from basic process models, and a new time series can be assigned to the most suitable model class. Typical models include autoregressive [22] and hidden Markov models [23]. The autoregressive model predicts the variable of interest by using a linear combination of the past values of the variable. It can flexibly handle various types of time series patterns. Hidden Markov models are widely used for time series and longitudinal data analysis because of their combination of (relative) simplicity and flexibility in adapting to new situations. Because of their flexibility, hidden Markov models are used for a wide variety of time series types, from continuous value, cyclic, and multivariate series to binary data, limited, and unlimited counts, and categorical observations.

Machine-learning-based approaches include convolutional neural networks (CNNs) and recurrent neural networks (RNNs) [24,25]. CNNs have been very successful in the past few years, demonstrating a cutting-edge performance compared with other approaches for some datasets in time-series classification problems. In addition, as a new deep architecture has been proposed for CNN, it has been revolutionarily used not only in time-series classification but also in the field of computer vision [26]. RNN is specialized in time-series data processing and has the advantage of being able to learn by simultaneously using input data and the data output from the previous step. Hochreiter and Schmidhuber [27] proposed the long short-term memory technique to solve the long-term dependence problem of an RNN. This technique is particularly advantageous for predicting time-series data, as it can prevent the inherent vanishing gradient problem of RNNs [28]. However, when dealing with large amounts of data, this approach requires a significant amount of computation time and memory size to optimize the learning process.

The abovementioned approaches are widely used in various fields, such as multimedia, healthcare, and finance [29]. These approaches have been applied in major research topics including earthquake prediction [30], terrestrial ecosystem dynamics [31], stock-price data, exchange-rate analysis [32], and bioinformatics [33]. However, they have not been specialized for binary time-series. Considering that time-series data are often transformed and analyzed through binarization in several fields, an analysis algorithm specialized for binary time-series still needs to be developed. The binarization process plays an important role in simplifying the analysis process while preserving the characteristics of the system. As an example of research related to the binarization process, Kedem and Fokianos [34] showed that the eruption period of the Old Faithful geyser in Yellowstone National Park can be binarized using threshold values. If the eruption lasted longer than 3 min, the eruption period was coded with a value of one, and for shorter eruptions, it was coded as zero. Bellégo [35] constructed a binary time-series by defining zero and one for recession and economic growth every two weeks, respectively, to analyze Italy’s economic dynamics. Adaes and Pires [36] constructed a binary time-series that defined ones and zeros when fine dust concentrations were above or below certain thresholds, respectively, to analyze the pollution kinetics of European cities.

In this study, we proposed a new approach based on the branch length similarity (BLS) entropy profile, which can capture the characteristic features of a binary time-series. We explored the local maximum (minimum) point, slope, and inflection point of the BLS entropy profile as the characteristic features of the binary time-series. In addition, to examine the applicability of the features, we compared the crawling trajectories of *Caenorhabditis elegans* exposed to two toxic substances (benzene and formaldehyde) using the local maximum point. In the discussion section, we briefly mention the problems that need to be explored mathematically in relation to the features proposed in this study and propose candidates for additional features based on the BLS entropy profile.

## 2. Materials and Methods

### 2.1. BLS Entropy and Its Profile

BLS entropy is observed in simple networks consisting of a node and several branches connected to the node [37,38,39] (Figure 1). The ratio of the length of each branch to the sum of the lengths of all branches is defined as the probability of each branch, as follows:(1)$$p_{j}=L_{j}/\sum_{k=1}^{n}L_{k},$$
where *n* is the number of branches in the network, and *L_k_* represents the length of the *k*th branch (*k* = 1, 2, 3,…, *n*). Thus, the BLS entropy can be mathematically written as:(2)$$S=-\sum_{j=1}^{n}p_{j}\log(p_{j})/\log(n)$$

The higher the similarity of the values of all the branches of the network, the closer the entropy (*S*) is to 1.0, and the lower the similarity, the closer it is to 0.0 [18]. For clarity, examples are provided in which the difference in length between branches is relatively large or small. When the length difference was large, the *S* value was low, and vice versa.

### 2.2. Time Circle for a Time Series

We introduce the concept of a *time circle* to calculate the BLS entropy value for the signal “1” in the binary time-series. A binary time-series is a sequence of “1” or “0” signals on the discrete time axis. In Figure 2, the upper image represents a binary time-series consisting of 400 random distributed “1” signals. We connected the first and last signals of the binary time-series by sequentially mapping the “1” signals to the circumference of a circle. We refer to this circle as the *time circle*. Without the time circle, it would be difficult to find a physical quantity that corresponds to the branch length of the BLS entropy in a time series. If the distance between signals in a time series is defined as the branch length, several signals that are very far apart from each other converge the BLS entropy value to zero, regardless of the distribution of the entire signal. In other words, the time circle prevents the information on the entire structure of the time series from being diluted by distant signals. After a time circle is created for a binary time-series, we can calculate the BLS entropy value for a signal by concatenating the signal with all the other signals. Then, we can obtain the entropy profile by sequentially calculating the entropy values for the signals along the direction of the time flow (bottom image of Figure 2).

## 3. Results

### 3.1. Characteristic Features of Binary Time-Series Appearing in the BLS Entropy Profile

To understand the binary time-series structure, we explored the local maximum (minimum) point, slope, and inflection point of the BLS entropy profile as the characteristic features of the binary time-series. To this end, we created a binary time-series *Q*(*t*) of length *L* (= 10,000) consisting of “1” signals heterogeneously distributed on the time axis. Using the neutral theory [39,40], we generated several heterogeneous landscapes determined by the control variable *H*, ranging from 0.0 to 1.0 (Figure 3A, left). The closer the *H* value is to 0.0, the sharper the peaks in the landscape, and the closer it is to 1.0, the smoother the peaks. The heights of the peaks in the landscape have a value between 0.0 and 1.0. Then, we assigned the values of 1 and 0 to grid sites where the height of the landscape was above and below 0.5, respectively (Figure 3A, right). Yellow and dark blue represent grid sites with the values of 1 and 0, respectively. Next, we created a single vector of length 10,000 by sequentially concatenating the column vectors of the binary image. This vector is a binary time-series *Q*(*t*) with the heterogeneity *H* (Figure 3B). Finally, we obtained the BLS entropy profile *S*($\tilde{t})$ from the time circle for *Q*(*t*) (Figure 3C). Here, *t* represents all the time values on the time axis (*t* = 1, 2,...), and $\tilde{t}$ is an array of *t* values where the signal “1” is located. Therefore, the length of $\tilde{t}$ is always less than or equal to *t*. We subjectively selected four interesting domains in *S*($\tilde{t}$) and investigated their corresponding domains in *Q*(*t*). *S*($\tilde{t}$) and *Q*(*t*) are represented by squares of the same color. Within the blue square of *S*($\tilde{t}$), the BLS entropy value gradually increases, whereas the signal density in the domain in *Q*(*t*) slowly decreases. When the inside of the blue square of *Q*(*t*) was enlarged, bands composed of bundles of “1” signals appeared, and the band length tended to decrease slightly along the time axis (see the top of Figure 3D). This reflects the fact that high (low) BLS entropy values indicate low (high) signal densities in the binary time-series. We prove this mathematically in Appendix A. In the red square of *S*($\tilde{t}$), the slope of the entropy profile is almost zero, and the BLS entropy value is relatively low. The red square of *Q*(*t*) has a relatively high signal density and, when enlarged, shows an almost uniform signal distribution (see the bottom of Figure 3D). Referring to the proof in Appendix A, the red square should show a tendency of decrease and then increase in the signal density; however, it shows a uniform trend. This is because the slope values of the entropy profile are relatively low. The black square of *S*($\tilde{t}$) contains the inflection point. The black square of *Q*(*t*) is in contact with the domain with a relatively high signal density on the left. Therefore, the position of the inflection point in *S*($\tilde{t}$) can be inferred as the time (*t* = *τ*) at which a significant change in the signal density occurs in *Q*(*t*). A significant change indicates that the degree of increase or decrease in the slope of the BLS entropy profile changes. The green square of *S*($\tilde{t}$) indicates the domain with moderate entropy values and low slopes of the entropy profile. The green square of *Q*(*t*) shows a signal distribution pattern with properly mixed characteristics of the blue and red squares.

We investigated the BLS entropy profile for a binary time-series with a simpler structure to facilitate a better understanding of the features. We created three uniform binary time-series, *Q_j_*(*t*) = 1 (*j* = 1, 2, 3), with *L* = 500 and different signal densities: *t* = {1, 3, 5,...} for *j* = 1, *t* = {1, 6, 11,...} for *j* = 2, and *t* = {1, 11, 21,...} for *j* = 3. We combined *Q*_1_(*t*) and *Q*_2_(*t*), as well as *Q*_1_(*t*), *Q*_2_(*t*), and *Q*_3_(*t*) to create two binary time-series, *Q*_12_(*t*) and *Q*_123_(*t*), with *L* = 1000 and 1500, respectively (Figure 4A). Therefore, the value of *τ* for *Q*_12_(*t*) is 500, and the values of *τ* for *Q*_123_(*t*) are 500 and 1000. *S*($\tilde{t}$) for *Q*_12_(*t*) has an inflection point at $\tilde{t}$ = 26, which exactly corresponds to *t* = 500, whereas it has a local maximum and minimum at $\tilde{t}$ = 13 and 75, respectively (Figure 4B). The two values correspond to *t* = 225 and 725 for *Q*_12_(*t*). Both values are near the center of *Q*_1_(*t*) and *Q*_2_(*t*). The inflection points for *Q*_123_(*t*) are at $\tilde{t}$ = 16 and 126, which exactly correspond to *t* = 500 and 1000, respectively. The local maximum and minimum values of *S*($\tilde{t}$) correspond to those near the central position on *Q*_1_(*t*), *Q*_2_(*t*), and *Q*_3_(*t*). As the BLS entropy profile is obtained by correlating all the signals in a binary time-series, the partial structure of the entropy profile contains information about the entire time series [39]. Therefore, the local maximum (minimum) point may deviate from the center point to some extent, depending on the overall signal distribution.

With conventional methods, determining the value of *τ* (the position of the inflection point in *S*($\tilde{t}$)) is difficult for a time series in which the signal density changes gradually and continuously. For example, let us consider a single *Q*(*t*) generated by combining two binary time-series *Q*_1_(*t*) = 1 for *t* = {1, 2, 4, 7,…, 497} and *Q*_2_(*t*) = 1 for *t* = {500, 501, 503, 506,..., 996} (Figure 4C). *Q*(*t*) shows a tendency of a gradual decrease and then increase again for the signal density. Here, to determine the value of *τ*, we should create a small time window on *Q*(*t*) and calculate the signal density within the window while shifting it in the direction of the time flow. Then, we need to find the location of the window where the density reaches the threshold set as the *τ* value. With this approach, the value of *τ* depends on the window size and threshold value. Alternatively, our approach can determine the value of *τ* by simply determining the inflection point of the BLS entropy profile for *Q*(*t*) (Figure 4C, left). The inflection points of *S*($\tilde{t}$) are $\tilde{t}$ = 15 and 76 (the right of Figure 4C), which are marked on *Q*(*t*) by two red lines. When differentiating *S*($\tilde{t}$) to find the inflection point, two peaks are observed at $\tilde{t}$ = 44 and 46 for *S*($\tilde{t}$). This is because the lengths of the two BLS entropy profiles corresponding to *Q*_1_(*t*) and *Q*_2_(*t*) are not the same.

In Figure 5, we examine the features of a binary time-series with *H* = 0.3. The local maximum (minimum) points (indicated by black triangles) and inflection points (indicated by red triangles) can be observed in the BLS entropy profile. The corresponding positions in *Q*(*t*) for the triangle positions are indicated by lines sequentially for each color. The results showed that a feature of BLS entropy sensitively detects moments of change in the signal density in a binary time-series visually.

### 3.2. Application: Characterization of Crawling Trajectories of Caenorhabditis Elegans 

Caenorhabditis elegans has 302 neurons, and their connections are well known. In addition, the worms have a transparent body, and are hence easy to observe with the eye. Because of these characteristics, they have been widely used for exploring the relationship between neural control and biomechanics in organisms [41,42]. This relationship has been revealed by analyzing the crawling behavior on agar or swimming behavior in water [43]. The analyses require algorithms to characterize behavioral trajectories.

In this study, we converted the trajectory of *C. elegans* into a binary time-series and quantified the trajectory using the characteristic features of the BLS entropy profile. To this end, we conducted experiments on the behavior of 30 wild-type adult *C. elegans* on agar. The worms were cultured in a Petri dish (60 mm in diameter, 15 mm in height) filled with nematode growth medium in an incubator at 20 °C. The OP50 strain of *E. coli* was used as food for the worms. The test worms were allowed to acclimate for 15 min before recording their behavior. We performed 10 replicate experiments with different individuals for a control group, a group exposed to 0.5 ppm benzene, and a group exposed to 0.5 ppm formaldehyde. The crawling activity of the worms was monitored using a Sony digital camcorder mounted vertically over 20 min at a frame resolution of 1/24 s. We extracted the central coordinate point of the worm body from each frame of the recorded movie. The coordinate points were used as the trajectory of the crawling movement. Figure 6 shows the typical trajectories of the worms belonging to the control, benzene-treated, and formaldehyde-treated groups. We measured the angle formed by the movement direction of the worm at times *t* and *t* + 1 to convert the two-dimensional trajectory into a binary time-series. The angle was converted to a binary number depending on the category to which it belongs. Here, the plane in which the individual moves is divided into eight angular categories, 0°–45°, 45°–90°,..., 315°–360°, and binarized into (001), (010),..., and (111), respectively. For example, if the worm advances by changing its direction by 25° with respect to the movement direction at time *t*, we can express the angle at time *t*+1 as the binary number, “001.” The trajectories of the control group tended to be complex in a long time scale and relatively simple sinusoidal in a short time scale. The trajectories of the benzene-treated group showed a simple movement in both the long and short time scales compared with those of the control group. On the other hand, the trajectories of the formaldehyde-treated group were simpler than those of the control group and slightly more complicated than those of the benzene-treated group. The trajectories appeared to be somewhat differentiated for each group based on the number of local maximum points in the BLS entropy profile. The “findpeaks” function supported by MATLAB (MathWorks, 2019) was used to find the number of local maximums. This function finds only those peaks that have heights above a certain threshold. When the threshold value was too low, too many peaks were found, whereas when the threshold value was too high, only a few peaks were found. We tested several thresholds and chose an appropriate value (0.01). For the BLS entropy profiles for the control, benzene-treated, and formaldehyde-treated groups, the numbers of local maximum points were (mean, standard deviation) = (8.25, 3.15), (3.87, 3.75), and (7.0, 3.10), respectively.

The trajectories of the control group were not statistically different from those of the formaldehyde-treated group, whereas they were significantly different from those of the benzene-treated group. In addition, there was no statistical difference between the trajectories of the benzene-treated and formaldehyde-treated groups (one-way ANOVA and Scheffe post-test, *p* < 0.05). 

In this application case, we showed that the features defined in the BLS entropy profile for a binary time-series can be effectively used through an appropriate binarization process even when analyzing non-binary time-series data, such as the behavior trajectory of an organism.

## 4. Discussion

In this study, we introduced the concept of a *time circle*, which allows the calculation of the BLS entropy profile from a binary time-series. Using the time circle, we observed that the local maximum (minimum) point, slope, and inflection point of the BLS entropy profile indicate the time at which the rate of change in the signal density becomes zero, the degree of change in the signal density, and the time at which the change in the signal density begins, respectively. In addition, through an application example, we showed that the findings are applicable to problems in various fields. In this section, we propose some candidates that capture characteristic features of a binary time-series.

The entropy profiles for *Q*_2_(*t*) and *Q*_3_(*t*), shown in Figure 7, contain several peaks. We observed that the peaks appeared when the distance between adjacent signals was suddenly lengthened and shortened. Understanding the peak generation in relation to the threshold would provide insights into a feature that can sensitively detect abnormal signals.

No-signals, that is “0” signals between a signal “1” and its neighboring signal “1,” cause a stepped structure in the BLS entropy profile. This structure is shown in the entropy profile in Figure 3C. Through a quantitative analysis of this effect, we selected the height as a feature that detects a specific no-signal distribution. Figure 6 shows that the BLS entropy profile for the movement trajectory of *C. elegans* tends to increase or decrease in the long and short time scales. We can use the Fourier transform algorithm to separate the high and low frequencies and analyze the tendency at different time scales separately, which could allow an elaborate classification of more sophisticated time-series structures. 

Another interesting feature is the similarity between two binary time-series based on the BLS entropy profile. Given a binary time-series *Q*_1_(*t*), we can create a time series *Q*_2_(*t*) by adding several signals close to each other to *Q*_1_(*t*), and another time series *Q*_3_(*t*) by adding the same number of signals scattered from each other to *Q*_1_(*t*). ED-based similarity indicates that the similarity between *Q*_1_(*t*) and *Q*_2_(*t*) and the similarity between *Q*_1_(*t*) and *Q*_3_(*t*) are the same. In other words, the ED does not reflect information on signal distribution. A new similarity (*ρ*) that overcomes this drawback can be defined as the correlation coefficient between the entropy profiles as follows:(3)$$\rho =\max\left\{\mathrm{corr}\left(S_{1}\left(\tilde{t}\right),\;shifted(S_{2}\left(\tilde{t}\right))\right)\right\},$$
here, “corr” represents the correlation coefficient between the two BLS entropy profiles, $S_{1}\left(\tilde{t}\right)$ and $S_{2}\left(\tilde{t}\right)$. “*shifted*
$\left(S_{2}\left(\tilde{t}\right)\right)$” shifts $S_{2}\left(\tilde{t}\right)$ cyclically by one time step. “max” determines the largest value among the calculated correlation coefficients. As the BLS entropy profile contains information about the relative distances between all “1” signals in the time series, *ρ* is significantly different from the existing distance-based similarity. Figure 7 shows the binary time-series, *Q*_1_(*t*), *Q*_2_(*t*), and *Q*_3_(*t*) for *H* = 0.0, 0.3, and 1.0, respectively, and their BLS entropy profiles. We compared the ED and ρ values for the two pairs of time series. In the ED, the similarity between *Q*_2_(*t*) and *Q*_3_(*t*) was the highest, followed by the similarity between *Q*_1_(*t*) and *Q*_3_(*t*) and the similarity between *Q*_1_(*t*) and *Q*_2_(*t*), in that order. However, the similarity between *Q*_1_(*t*) and *Q*_2_(*t*) visually appears to be the highest. The ρ measurement showed that the similarity between *Q*_1_(*t*) and *Q*_2_(*t*) was the highest, followed by the similarity between *Q*_2_(*t*) and *Q*_3_(*t*) and the similarity between *Q*_1_(*t*) and *Q*_3_(*t*), in that order. Even through a visual comparison, our similarity has an advantage over the existing distance-based similarity. In addition to *ρ*, which we used, Ultrametric distance measurement based on the Pearson linear correlation coefficient and Menhattan distance measurement can be used [44]. It would be interesting to study to determine an appropriate correlation coefficient according to the structure of the time series.

It is worthwhile to propose the concept of the time circle, which allows the calculation of the BLS entropy value for a binary time-series and shows that the characteristic features of the time series can be captured from the BLS entropy profile. In addition, we believe that various features can be defined based on the BLS entropy profile, and that they can be effectively applied to various problems expressed in binary time-series.

## Figures and Tables

**Figure 1 entropy-23-00480-f001:**
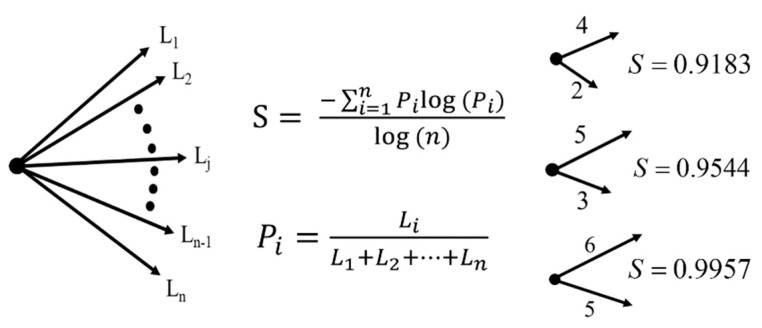
Definition of BLS entropy in a simple network composed of one node and several branches, and an example of BLS entropy change according to the change in the branch length.

**Figure 2 entropy-23-00480-f002:**
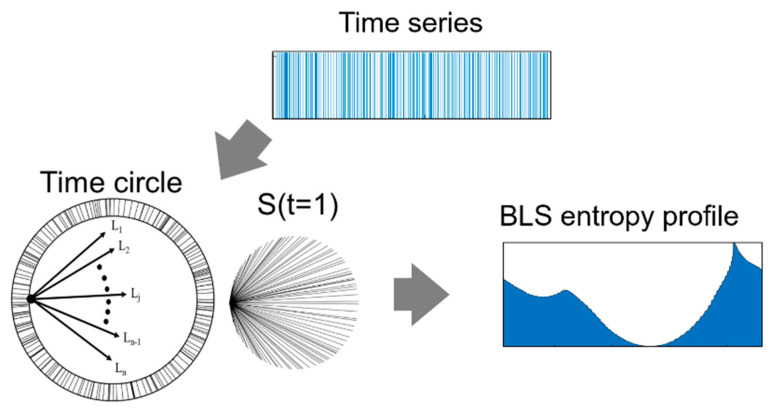
Time circle generated by mapping signals of random binary time-series with length *L* = 400 on the circumference of a circle and BLS entropy profile obtained from the “1” signals on the time circle.

**Figure 3 entropy-23-00480-f003:**
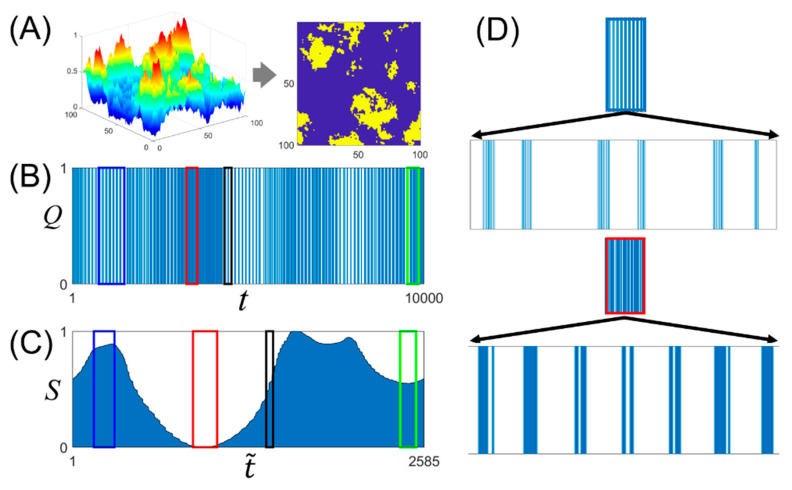
(**A**) Heterogeneous landscape according to the control variable, *H*, and its binary images through the binarization process, (**B**) binary time-series for the image, and (**C**) BLS entropy profiles corresponding to the time series. (**D**) Signal pattern with the enlarged blue square (upper figure) and red square (lower figure) on the BLS entropy profile.

**Figure 4 entropy-23-00480-f004:**
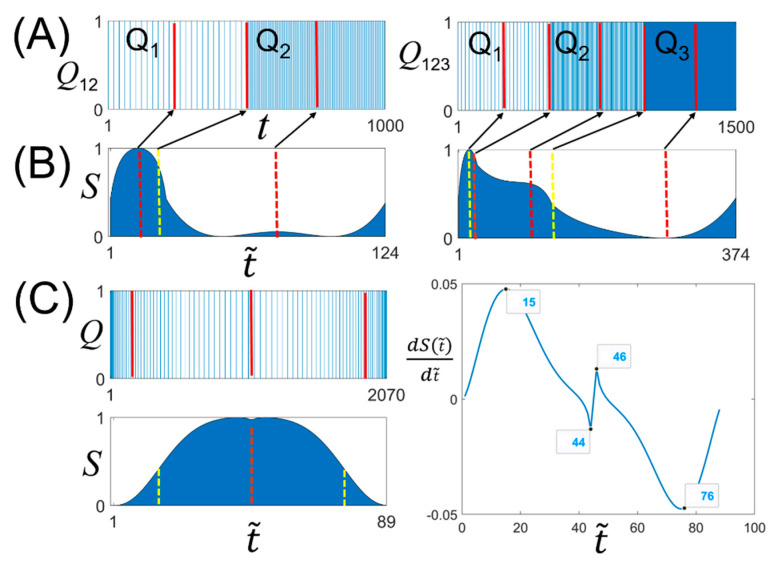
(**A**) Two binary time-series, *Q*_12_(*t*) and *Q*_123_(*t*), created by combining *Q*_1_(*t*), *Q*_2_(*t*), and *Q*_3_(*t*). *Q*_12_(*t*) was created by the sequential combination of *Q*_1_(*t*) and *Q*_2_(*t*), and *Q*_123_(*t*) was generated by the combination of *Q*_1_(*t*), *Q*_2_(*t*), and *Q*_3_(*t*). (**B**) BLS entropy profiles corresponding to *Q*_12_(*t*) and *Q*_123_(*t*), and (**C**) binary time-series, *Q*(*t*), in which the signal density linearly decreases and then increases, its BLS entropy profile, and the derivative function of the entropy profile ($S\left(\tilde{t}\right)$).

**Figure 5 entropy-23-00480-f005:**
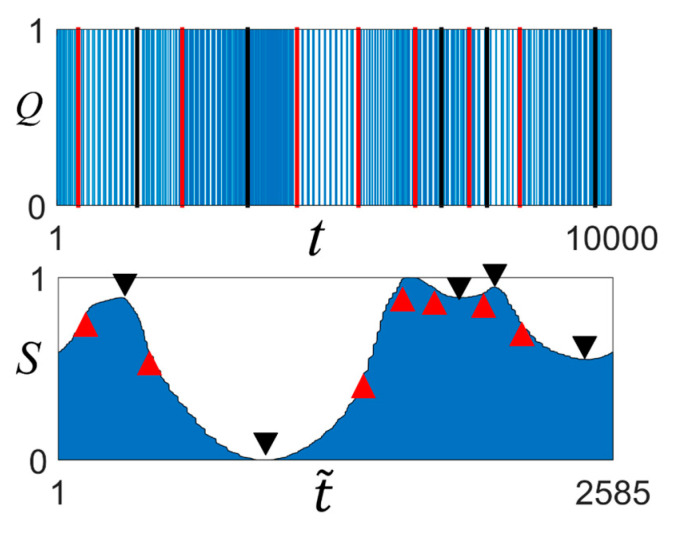
Heterogeneous binary time-series with *H* = 0.3, *Q*(*t*), and its local maximum (minimum) points (marked by a black triangle) and inflection points (marked by a red triangle) on the BLS entropy profile for the time series.

**Figure 6 entropy-23-00480-f006:**
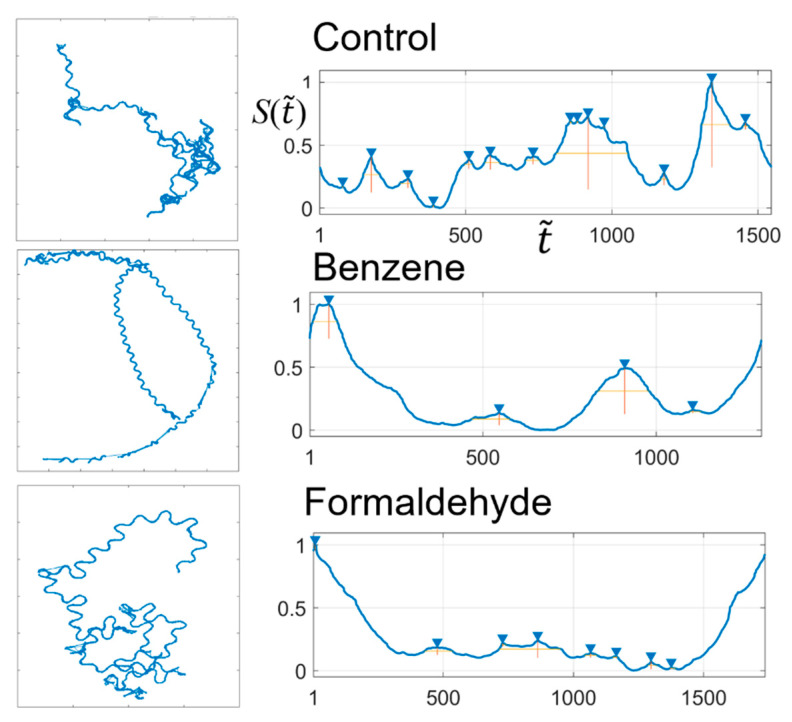
Crawling trajectories of *C. elegans* for the control, benzene-treated (0.5 ppm), and formaldehyde-treated (0.5 ppm) groups, and BLS entropy profiles for the trajectories. The triangles represent the locations of the local maximum points on the entropy profile.

**Figure 7 entropy-23-00480-f007:**
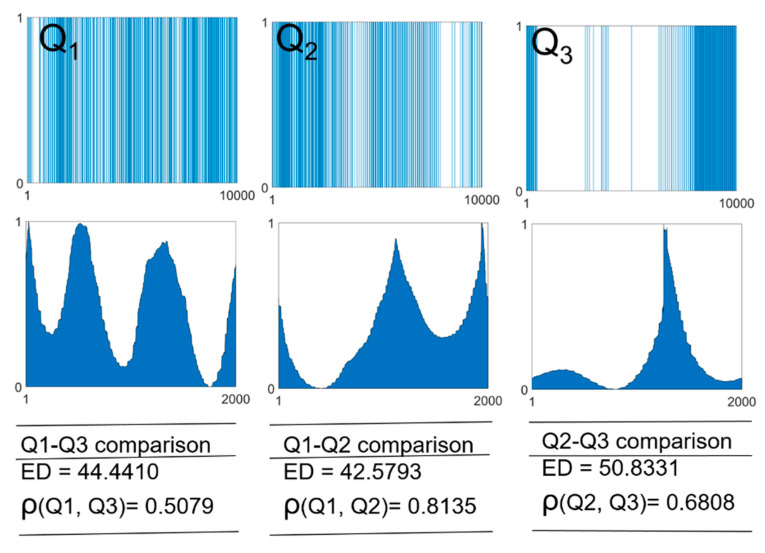
Binary time-series consisting of heterogeneously distributed “1” signals, *Q*_1_(*t*) (*H* = 0.1), *Q*_2_(*t*) (*H* = 0.2), *Q*_3_(*t*) (*H* = 0.3), and their BLS entropy profiles. Here, *H* indicates the heterogeneity of the binary time-series. ED and *ρ* represent the similarity based on the ED and the similarity based on the BLS entropy profile, respectively.

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
