# Peer review of "Novel Features for Binary Time Series Based on Branch Length Similarity Entropy"

_entropy, 2021, doi:10.3390/e23040480_

Round 1

Reviewer 1 Report

The article contains original materials and interesting considerations which are consistent with the pattern of research. Solid methodology of the research with statistical analysis.

Therefore contribution to existing knowledge is appreciated ; the research report is well organized but not easily readable, due to too many Englsih style and grammar ‘’approximations’’

Conclusion : I cannot find much weakness in the scientific content of the article, but the English has to be much improved.

A few details : I cannot list many ‘’style, content, grammatical, approximations’’, but for example, consider that the tense of verbs should be completely rechecked ; the present tense should always been preferred. At once : Abstract : « we showed » ; nooooo ; you show now, thus « we show », etc.etc.

Also the vocabulary is defectuous.

Abstract and thereafter : what is a ‘’binary time series’’ ?

Fig. 6. Benzen, should be Benzene !

Line 295 : « preliminary study » ; what does that mean ?

Line 298 : « would provide » ; why don’t you do it ?

Line 302 : « we could select » ; do you mean « we selected » ?

Line 310 : « we can create » : did you see a paper by Petroni et al. (Petroni, F., Ausloos, M., & Rotundo, G. (2007). Generating synthetic time series from Bak–Sneppen co-evolution model mixtures. Physica A: Statistical Mechanics and its Applications, 384(2), 359-367.) on combining time series. IN fact, see also (about Line 31) on distance between time series : MiÅ›kiewicz, J. (2012). Analysis of time series correlation. The choice of distance metrics and network structure. Acta Phys. Pol. A, 121, B-89.

Reviewer 2 Report

This study provides a branch length similarity (BLS) entropy characterization for binary time series, and with an interesting example of  C. elegans movement. Overall I enjoyed reading the manuscript. The description is generally clear and sound. My concerns are:

It seems Fig. 2 and Fig. 3 are redundant. Maybe the author mistakenly used the same figure twice.  

I still have trouble comprehending what a branch actually represent in the tree. For instance, in Fig. 1, are L1 and L2 successive branches of consecutive 1s?

The authors discussed the novelty of BLS entropy but why do we need this new feature especially for binary time series? The authors did not address how it compare with existing time series/signal features. I would like to see the real world comparison in the C.elegans example.
